# Prognostic role and biologic features of Musashi-2 expression in colon polyps and during colorectal cancer progression

Leonid Kharin[1,2]*, Igor Bychkov[2]*, Nikolay Karnaukhov[1], Mark Voloshin[3], Rushaniya Fazliyeva[2], Alexander Deneka[2], Elena Frantsiyants[1], Oleg Kit[1], Erica Golemis[2], Yanis Boumber[2,4,5]

1 National Medical Research Center of Oncology, Rostov-on-Don, Russian Federation, 2 Molecular Therapeutics Program, Fox Chase Cancer Center, Philadelphia, Pennsylvania, United States of America, 3 Rostov State Medical University, Rostov-on-Don, Russian Federation, 4 Kazan Federal University, Kazan, Russian Federation, 5 Department of Hematology/Oncology, Section of Thoracic Oncology, Fox Chase Cancer Center, Philadelphia, Pennsylvania, United States of America

* kharinleonid@gmail.com (LK); igor3mkk@gmail.com (IB)

## Abstract

### Background

The RNA-binding protein Musashi-2 (MSI2) controls the translation of proteins that support stem cell identity and lineage determination and is associated with progression in some cancers. We assessed MSI2 as potential clinical biomarker in colorectal cancer (CRC) and tubulovillous adenoma (TA) of colon mucosa.

### Methods

We assessed 125 patients, of whom 20 had polyps of the colon (TAs), and 105 had CRC. Among 105 patients with CRC, 45 had stages I-III; among metastatic CRC (mCRC) patients, 31 had synchronous and 29 metachronous liver metastases. We used immunohistochemistry to measure MSI2 expression in matching specimens of normal tissue versus TAs, primary CRC tumors, and metastases, correlating expression to clinical outcomes. We analyzed the biological effects of depleting MSI2 expression in human CRC cells.

### Results

MSI2 expression was significantly elevated in polyps versus primary tissue, and further significantly elevated in primary tumors and metastases. MSI2 expression correlated with decreased progression free survival (PFS) and overall survival (OS), higher tumor grade, and right-side localization (p = 0.004) of tumors. In metastases, high MSI2 expression correlated with E-cadherin expression. Knockdown of MSI2 in CRC cells suppressed proliferation, survival and clonogenic capacity, and decreased expression of TGFβ1, E-cadherin, and ZO1.

**Data Availability Statement:** All relevant data are within the manuscript and its Supporting information files.

**Funding:** The authors were supported by the National Institutes of Health (NIH) R21 CA223394 and R01 CA218802 grants (to Y.B.), NIH CA228187 (to EAG), by the Russian Government Program for Competitive Growth of Kazan Federal University (to AD and YB), and by the NIH Core Grant CA006927 (to Fox Chase Cancer Center).

**Competing interests:** No authors have competing interests.

## Conclusion

Elevated expression of MSI2 is associated with pre-cancerous TAs in the colonic mucosa, suggesting it is an early event in transformation. MSI2 expression is further elevated during CRC progression, and associated with poor prognosis. Depletion of MSI2 reduces CRC cell growth. These data imply a causative role of MSI2 overexpression at multiple stages of CRC formation and progression.

## Introduction

Despite the development of new methods of screening, diagnosis and treatment, colorectal cancer (CRC) continues to be a significant source of cancer mortality worldwide, both in men and women [1]. Worldwide, in 2018, 1.8 million people were diagnosed with CRC and more than 850,000 died of this disease [2]. The advent of targeted therapy and the use of a multidisciplinary approach to treatment have improved overall survival (OS) [3]. The 5-year relative survival rate for all CRC stages has improved from 50% in the 1970s to 64% during the 2010s, and in patients diagnosed with distant metastases, the 2-year survival rate improved from 21% during the mid-1990s to 37% during 2010s [4,5]. However, mCRC remains a major cause of death in cancer patients.

Up to 21% of patients are diagnosed with stage IV cancer, with typical OS of 7.2 months following diagnosis [6]. Liver metastases are the most common sites of CRC metastasis. At the initial diagnosis, liver metastases are detected in 15–25% of patients (synchronous metastases). For another 18–25% of patients diagnosed at stages I–III [7], liver metastases are observed within 5 years after diagnosis (metachronous metastases) [8]. Resistance to therapy and subsequent mortality in mCRC is associated with many biological changes in the tumor [9]. A critical question is the degree to which patients likely to develop an aggressive disease course, including metachronous metastases, can be identified at an earlier disease stage. Such early identification would allow these individuals to receive more active surveillance and potentially nominate them for more aggressive treatment. Hence, there is much interest in defining biomarkers of aggressive malignancy in early stages of CRC, and in premalignant colon mucosa [9,10].

The RNA-binding protein Musashi-2 (MSI2) controls the translation of proteins that support stem cell identity and lineage determination [11], and is a potential biomarker for CRC prognosis. Elevated expression of MSI2 plays an important role during progression [12], dissemination [13], and drug resistance [13,14] in numerous solid [15–18] and hematological malignancies [19–21]. Importantly, elevated MSI2 expression at an early tumor stage may predict worse outcomes in later tumors. For example, in non-small cell lung cancer (NSCLC), elevated MSI2 expression was first detectable in early stage tumors, and continued to rise during tumor progression [12]. Mechanistically, MSI2 was found to govern expression of proteins in the TGF-β signaling pathway, which is known to undergo a transition from growth-inhibiting to invasion-promoting during oncogenesis [22], supporting the process of epithelial-mesenchymal transition (EMT). These data suggested MSI2 may play an important role in this shift towards invasive and metastatic tumors [23].

Some work has suggested MSI2 regulation of genes governing EMT and aggressive tumor growth may be relevant in CRC. In studies using multiple mouse models for CRC formation, MSI2 overexpression was shown to inhibit expression of tumor suppressors, partially phenocopying changes induced by loss of *APC*. Moreover, loss of Musashi function in transgenic

mouse model compromised the growth of CRCs, and influenced response to investigational targeted therapies in pre-clinical models [24,25]. Based on these data, we hypothesized that MSI2 expression may provide a useful biomarker for tumor aggressiveness in CRC. In this study, using the histological material of patients with tubulovillous adenomas (Tas) of the colon and patients with verified CRC, we evaluated the expression of MSI2 and its effect on clinical characteristics. In addition, we assessed the role of MSI2 in regulating the proliferative and colony forming ability of human CRC cell lines, and whether MSI2 regulated the expression of proteins associated with EMT and tumor stage, including E-cadherin (CDH1), the tight junction protein ZO1, and the cytokine TGFβ1.

## Patients, methods, and materials

### Patients

The patient group consisted of 66 men and 59 women with an average age of 59 years (range 31–78 years). Of these 125 patients, 20 had colon tubulovillous adenomas (TAs, or colon polyps), and 105 had CRC (53 patients with left-sided and 52 patients with right-sided tumors). Patients with polyps and CRCs were matched for age, sex and absence of comorbidity. Of the CRC patients, 45 had locoregional cancer (stages I-III), and 60 patients had mCRC with liver metastases; of the specimens from mCRC patients, 31 patients had been diagnosed with synchronous and 29 with metachronous metastases.

### Sample collection, tissue preparation, immunohistochemistry and quantitative analysis

All patients with TA polyps or colon cancer in this study signed approved National Medical Research Center of Oncology (Rostov-on-Don, Russia) institutional consent forms (IRB No. 6, 02/03/2018) that allowed us to collect specimens, perform IHC, and analyze data. All samples were de-identified. No research activity was done prior to obtaining appropriate IRB permissions. Normal mucosa, colorectal polyps and tumors were collected. Tissues were collected and fixed in 10% phosphate-buffered formaldehyde (formalin) for 24–48 hrs, dehydrated and embedded in paraffin. Tissues were processed by dehydration through ethanol followed by xylene (70% ethanol, 3 hr; 95% ethanol, 2 hr; 100% ethanol, 2 hr; ethanol-xylene, 1hr; xylene, 3hr) then immersed in paraffin. Paraffin blocks were cut into 5 μm sections, mounted on microscope slides, and stored at room temperature until used. Prepared specimens were analyzed by hematoxylin and eosin (H&E) staining (SigmaAldrich, St. Louis, MO).

### Immunohistochemistry and assessment of data

IHC was performed by standard protocols specified by antibody manufacturers, using anti-MSI2 antibody (Abcam, cat. ab50829 with 1:200 dilution), anti-E-Cadherin (CDH1) antibody (Abcam, cat. ab1416 with 1:200 dilution) and secondary goat anti-rabbit IgG (Abcam, cat. N˚ ab205718 with 1:2000 dilution). Immunostained slides were scanned using an Aperio Scan-Scope CS scanner (Aperio, Vista, CA) and Vectra Automated Quantitative Pathology Imaging System (Perkin Elmer, Waltham, MA). Scanned images then were viewed with Aperio Image-Scope software. Selected regions of interest were outlined manually by a certified clinical pathologist (N. Karnaukhov). The percentage of cells at each staining intensity level was determined, and an H-score was assigned and calculated for each slide using the following formula: [1 × (% cells 1+) + 2 × (% cells 2+) + 3 × (% cells 3+)]. H-scores were subsequently used for analysis.

## Statistical analysis

For statistical processing, the quantitative indicators of MSI2 and E-cadherin IHC expression in patients' samples, evaluated by H-score method was approximated into qualitative ones by dividing the expression level into high and low by median in every type of tissue respectively. Obtained result were compared using the Kruskal-Wallis test. Categorical data were evaluated using the Pearson chi-square test, adjusted by Yates's correction for continuity. The correlation coefficients between protein expression and clinical and pathological parameters were estimated using Spearman correlations. A linear regression model was used to investigate correlation in MSI-2 and E-cadherin protein expression levels. The Kaplan-Meier method was used to construct survival curves, as well as to calculate the median of survival. Differences between groups were established using a log-rank test. All statistical analyzes were performed using SPSS 16.0 for Windows (SPSS, Inc., Chicago, Illinois, USA). All P values were two-sided, with $P < 0.05$ indicating a statistically significant difference.

## Cell culture

The RKO (ATCC CRL-2577) and HCT116 (ATCC CCL-247) colorectal carcinoma cell lines were obtained from the FCCC Cell Culture Facility and authenticated by genotyping performed by IDEXX BioResearch (Columbia, MO, USA). Both cell lines were cultured in RPMI-1640 media containing 10% fetal bovine serum (FBS), L-glutamine, non-essential amino acids, pyruvate and penicillin/streptomycin (pen/strep). For the list of cell line derivatives used in the study, see S1 Table.

## shRNA targeting sequences and lentivirus production

Short hairpin RNAs (shRNAs) were obtained from SIGMA-ALDRICH (St Louis, MO) (S2 Table). Two MSI2- targeting sequences were used:

MSI2-sh1:

```
CCGGGTGGAAGATGTAAAGCAATATCTCGAGATATTGCTTTACATCTTCCACTTTTTG
```

MSI2-sh2:

```
CCGGCCCAACTTCGTGGCGACCTATCTCGAGATAGGTCGCCACGAAGTTGGGTTTTTG
```

Empty vector Tet-pLKO-puro system (Plasmid #21915, Addgene, Cambridge, MA) was used as a control. To prepare lentivirus for introduction of shRNAs into CRC cells, HEK-293T cells were transfected with shRNA lentivirus prepared in the Tet-pLKO-puro system (Plasmid #21915, Addgene, Cambridge, MA), with the psPAX2 (Plasmid # 12260, Addgene, Cambridge, MA) and pMD2.G (Plasmid #12259, Addgene, Cambridge, MA packaging plasmids Pspax #12260 and pMD2.G # 12259, both from Addgene (Cambridge, MA). Media containing lentiviral particles was collected on day 4. Subsequently, CRC cells were infected with lentivirus and selected by growth in media, as described above, using standard methods [12].

## *In silico* evaluation of MSI2 binding to EGFR mRNA

Human genome sequences for *TGFβ-1*, *CDH1* and *ZO-1* were obtained from the UCSC Human Gene Sorter December 2013 (GRCh38/hg38) assembly, and scanned for Musashi binding motifs defined by Wang et al [25]: see S3 Table.

## mRNA expression

Total DNA-free mRNA was isolated from HCT116 (-pLko, sh1, sh2) and RKO (-pLko, sh1, sh2) cell lines using Quick-RNA™ MiniPrep (#R1054) (Zymo Research, Orange, CA), reverse-

transcribed using Moloney murine leukemia virus reverse transcriptase (Ambion-Thermo Fisher Scientific, Waltham, MA) with a mixture of anchored oligo-dT and random decamers used as primers (Integrated DNA Technologies, Coralville, IA). mRNA expression for MSI2 and other genes of interest were analyzed by quantitative RT-PCR, using the primers listed in S4 Table.

## Western blotting

For western blotting analysis, RKO and HCT-116 transfected with MSI2-targeting or control lentiviruses were plated and treated with 1μg/μl of doxycycline for 48 hours to induce MSI2 knockdown. Cells were then washed twice with cold PBS and were disrupted in CelLytic M lysis buffer (Sigma-Aldrich, St. Louis, MO) supplemented with protease and phosphatase inhibitor cocktails (Roche, Basel, Switzerland). Whole cell lysates were used directly for SDS–PAGE and Western blotting, using standard procedures. After centrifugation, supernatants were collected, and total protein quantified by BCA (Thermo Scientific). Cell lysates were separated by 4–15% SDS–PAGE and transferred to PVDF membrane. Membranes were blocked and were blotted overnight (4˚C) for TGβ-1 (ab92486, AbCam, 1:1000 dilution), ZO-1 (sc-33725, Santa Cruz; 1:1000 dilution), CDH1 (#3195, Cell Signaling; 1:1,000 dilution), MSI2 (ab76148, Abcam; 1:2,000 dilution) and β-actin (A3854, Sigma-Aldrich; 1:5000 dilution) used as primary antibodies. Secondary anti-mouse and anti-rabbit horseradish peroxidase–conjugated antibodies (GE Healthcare, Little Chalfont, UK) were used at a dilution of 1:10,000 for visualization of Western blots and blots developed by chemiluminescence using the West-Pico system (Pierce, Waltham, MA). Image analysis was done using ImageJ (National Institutes of Health, Bethesda, MD), with signal intensity normalized to β-actin or total level of detected proteins. Data was analyzed in Excel by paired t-test to determine statistical significance.

## Colony formation assay

For each test condition, 500 cells were plated in 6-well plates and incubated in complete media for 12 days. Cells were then fixed in 10% acetic acid/10% methanol solution and stained with 0.5% (w/v) crystal violet. A colony was defined as consisting of >50 cells, and counted digitally using ImageJ software. GraphPad software was used statistical analysis.

## Cell proliferation assays

Cells ($1–2 \times 10^3$/well) were plated in quadruplicate plates in RPMI1640 media with 10% FBS in 96-well cell culture plates for 5 days. On days 1–5, CellTiter-Blue® (Promega, Fitchburg, WI) or WST-1 reagent (Sigma-Aldrich, St. Louis, MO) reagent were added to wells one one plate; after 2 hours incubation at 37˚C, optical density readings were made to visualize percentage of live cells, using a Perkin-Elmer ProXpress Visible-UV-fluorescence 16-bit scanner (Perkin-Elmer, Waltham, MA). GraphPad software used for statistical analysis.

## TCGA analysis

The Cancer Genome Atlas (TCGA, data release from January 28 2016) results shown in this study are based upon Firehose legacy data generated by Broad Institute (https://gdac.broadinstitute.org/ (Version: 1.1.40)). Data for 83 protein samples and 203 matched RNA samples for colorectal adenocarcinoma were downloaded from www.cbioportal.org (Date: January 22 2020). We used internal www.cbioportal.org software (R/MATLAB) for creation of correlation graphs, and calculation of Pearson and Spearman correlation coefficients. Correlation

data for tumor samples from matching RNA and protein samples sets for the colorectal adeno-carcinoma study were downloaded from www.cbioportal.org [26,27].

## Results

### Expression of MSI2 based on tumor stage, grade, and survival

We compared immunohistochemical staining intensity for MSI2 in normal colon mucosa, polyps, and invasive colorectal carcinomas (CRCs) (Fig 1A). Based on analysis of H-score, there was a significant increase in MSI2 protein expression in colorectal polyps (all tubulovillous adenoma by histology) versus normal mucosa (median values, 140 versus 60; average values, 138.75 versus 60.25; p<0.001) (Fig 1B). We also compared MSI2-stained matched normal mucosa versus primary and metastatic tumor samples for 105 CRC patients, including 45 patients with localized (stage I-III) CRC and 60 patients mCRC (Fig 1C). In this case, MSI2 H-scores were significantly elevated in primary tumors versus normal mucosa (median values, 190 versus 70; average values, 197 versus 69.4; p<0.001). H-values for MSI2 were further elevated in tumor metastases (median values, 270 versus 190; average, 267.3 versus 197; p<0.001 for metastases versus primary tumor). High expression of MSI2 in the primary tumors correlated with differentiation grade, both using H-score (p<0.001) (Fig 1D) and percent MSI2 expression level (p = 0,037) methods (Table 1), but not with depth of invasion, lymph node positivity, tumor location (right or left sidedness), presence of distant metastases or tumor stage (Table 1).

We also assessed the relationship between MSI2 expression and the clinical and pathological features of liver metastases from the same patients for whom we had assessed primary tumors (Table 2). MSI2 protein expression was higher in the liver metastases of tumors arising on the right side (median values with quartiles 260 [245; 275] versus 280 [270;290]; average values, 258 versus 273; (p = 0.004). A high level of MSI2 expression in the metastasis was also associated with lymph node positivity (median values with quartiles 270 [250;280] for N0 versus 265 [242;280] for N1 versus 280 [275;290] for N2; average values, 268 versus 262 versus 281; p = 0.04), and with higher differentiation grade of the primary tumor (p = 0.000) (Table 2). In these tumors, there was also no correlation between the level of MSI2 expression and initial stage at diagnosis of tumors that subsequently metastasized (e.g. o metachronous mCRC) (p = 0.793) (Table 2). For both primary tumors and their metastases, no statistically significant relationship was identified for MSI2 expression with patient sex, age, or type of metastasis (synchronous or metachronous).

For each patient cohort, we evaluated progression free survival (PFS) and overall survival (OS) associated with MSI2 expression in the primary tumor, stratifying the level of MSI2 as high or low expression based on the median H-score for each group (Table 3). Based on this analysis, PFS was significantly lower in patients with high MSI2 expression in the primary tumors, whether patients were diagnosed with synchronous (p = 0.0045) or metachronous metastasis (p = 0.001) (Fig 1E). However, MSI2 expression level in the primary tumors of mCRC patients did not influence OS for these patients (Fig 1E and Table 3). Further, MSI2 expression level in localized CRC (stage I-III) did not correlate with OS or PFS (Fig 1E).

We also evaluated the relationship between OS and PFS and MSI2 expression level in the matched liver metastases for 60 mCRC patients. MSI2 expression above the median H-score in these liver metastases was strongly associated with poor OS (log-rank = 12.555; p <0.0001) and PFS (log-rank = 13.481; p <0.0001) (Table 3). Additional relationships were found between OS and diagnosis with synchronous liver metastasis (p = 0.003), the depth of invasion (p = 0.038), the presence of metastases in the lymph nodes (p<0.0001), and the grade of differentiation (p = 0.012) (Table 3).

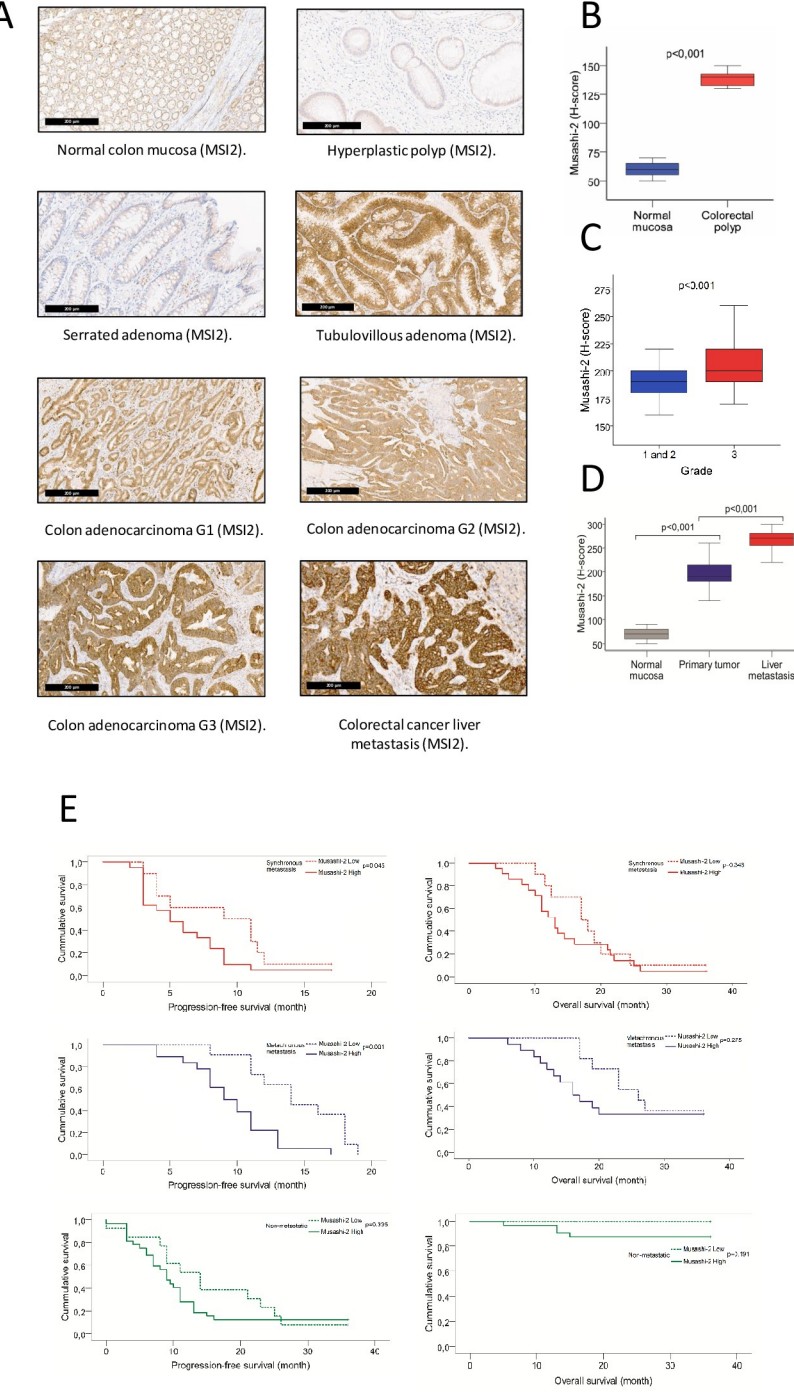

**Fig 1. An increased level of MSI2 expression is associated with colon polyps, primary tumor, liver metastasis and poor survival.** (**A**) Representative IHC images of MSI2 expression in normal colon mucosa, colon polyps, primary CRC, and CRC liver metastasis. (**B**) H-score for MSI2 expression in normal mucosa (n = 20) and colorectal polyps (n = 20). (**C**) H-score for MSI2 expression in normal colon mucosa, primary CRC tumor, and matched CRC liver metastasis (n = 60). (**D**) H-score for MSI2 expression in primary tumors with the grades of differentiation (n = 105). (**E**) Kaplan-Meier progression-free survival (PFS) and overall survival (OS) for patients with low (dotted line) and high (solid line) levels of Musashi-2 protein expression based on median values for H-score in primary tumor. Data is presented for patients diagnosed with stage I-III CRC (green), with synchronous metastases (stage IV) (red), or stage I-III but subsequently with metachronous metastases(blue), based on evaluation of primary tumors collected at initial diagnosis.

**Table 1. Comparison of the frequencies of qualitative indicators in primary colorectal tumors with low and high expression of MSI2 (n = 105).**

| Clinicopathological parameters | Total Number of patients | Musashi-2 protein expression | | χ2 | p-value | r |
|---|---|---|---|---|---|---|
| | | Low (n. (%)) | High (n. (%)) | | | |
| **Gender** | | | | | | |
| Male | 56 | 19 (34%) | 37 (66%) | 0.023 | 0.878 | -0.035 |
| Female | 49 | 15 (31%) | 34 (69%) | | | |
| **Age range** | | | | | | |
| ≤65 years | 54 | 16 (30%) | 38 (70%) | 0.169 | 0.681 | -0.061 |
| >65 years | 51 | 18 (35%) | 33 (65%) | | | |
| **Type (Localized/Metastasized)** | | | | | | |
| Non-metastatic | 45 | 13 (29%) | 32 (71%) | 0.659 | 0.719 | -0.078 |
| Synchronous | 31 | 10 (32%) | 21 (68%) | | | |
| Metachronous | 29 | 11 (38%) | 18 (62%) | | | |
| **Location of tumor** | | | | | | |
| Left-sided | 53 | 21 (40%) | 32 (60%) | 1.939 | 0.164 | 0.156 |
| Right-sided | 52 | 13 (25%) | 39 (75%) | | | |
| **Stage** | | | | | | |
| I | 13 | 4 (31%) | 9 (69%) | 2.542 | 0.468 | 0.051 |
| II | 22 | 10 (45%) | 12 (55%) | | | |
| III | 39 | 10 (25%) | 29 (75%) | | | |
| IV | 31 | 10 (33%) | 21 (67%) | | | |
| **Depth of invasion** | | | | | | |
| T2 | 17 | 6 (35%) | 11 (65%) | 0.118 | 0.943 | 0.033 |
| T3 | 68 | 22 (32%) | 46 (68%) | | | |
| T4 | 20 | 6 (30%) | 14 (70%) | | | |
| **Lymph node metastasis** | | | | | | |
| N0 | 35 | 14 (40%) | 21 (60%) | 2.065 | 0.356 | 0.139 |
| N1 | 51 | 16 (31%) | 35 (69%) | | | |
| N2 | 19 | 4 (21%) | 15 (79%) | | | |
| **Grade of differentiation** | | | | | | |
| High | 43 | 9 (21%) | 34 (79%) | **4.361** | **0.037** | **0.204** |
| Moderate and low | 62 | 25 (40%) | 37 (60%) | | | |
| **E-cadherin in primary tumor** | | | | | | |
| Low | 12 | 3 (25%) | 9 (75%) | 0.000 | 1.000 | -0.040 |
| High | 14 | 5 (32%) | 11 (68%) | | | |
| **E-cadherin in metastasis** | | | | | | |
| Low | 12 | 5 (42%) | 7 (58%) | 1.750 | 0.186 | 0.333 |
| High | 16 | 2 (12%) | 14 (88%) | | | |

Categorical data were evaluated using the Pearson χ2 test, adjusted by Yates's correction for continuity. The correlation coefficients between protein expression and clinical and pathological parameters were estimated using a Spearman correlation. Significant distinctions are indicated in bold font.

## Relationship between MSI2 and E-cadherin in CRC

Because our data suggested a potential relationship between MSI2 expression and tumor grade in CRC metastases, we investigated the expression of E-cadherin (CDH1) (Fig 2A), in relation to MSI2, in 28 randomly chosen patients with mCRC. Comparison of the medians of E-cadherin expression by H-score in the matched primary tumor and metastasis revealed statistically significant differences, with elevated expression of E-cadherin in metastasis ($p < 0.001$) (Fig 2B). For further statistical analysis, quantitative indicators of CDH1 expression in the

**Table 2. Comparison of the frequencies of qualitative indicators in CRC liver metastases with low and high expression level of Musashi-2 (n = 60).**

| Clinicopathological parameters | Total Number of patients | Musashi-2 protein expression | | χ2 | p-value | r |
|---|---|---|---|---|---|---|
| | | Low (n. %) | High (n. %) | | | |
| **Gender** | | | | | | |
| Male | 32 | 12 (37%) | 20 (63%) | 0.000 | 1.000 | -0.018 |
| Female | 28 | 10 (36%) | 18 (64%) | | | |
| **Age range** | | | | | | |
| ≤65 years | 22 | 5 (23%) | 17 (77%) | 2.036 | 0.154 | -0.220 |
| >65 years | 38 | 17 (45%) | 21 (55%) | | | |
| **Type (Localized/Metastasized)** | | | | | | |
| Synchronous | 31 | 13 (42%) | 18 (58%) | 0.369 | 0.543 | 0.113 |
| Metachronous | 29 | 9 (31%) | 20 (69%) | | | |
| **Location of tumor** | | | | | | |
| Left-sided | 25 | 15 (60%) | 10 (40%) | **8.399** | **0.004** | **0.409** |
| Right-sided | 35 | 7 (20%) | 28 (80%) | | | |
| **Initial stage of CRC** | | | | | | |
| I | 5 | 2 (40%) | 3 (60%) | 1.036 | 0.793 | -0.055 |
| II | 6 | 2 (33%) | 4 (67%) | | | |
| III | 18 | 5 (28%) | 13 (72%) | | | |
| IV | 31 | 13 (42%) | 18 (58%) | | | |
| **Depth of invasion** | | | | | | |
| T2 | 7 | 2 (29%) | 5 (71%) | 0.878 | 0.645 | 0.033 |
| T3 | 42 | 17 (41%) | 25 (59%) | | | |
| T4 | 11 | 3 (27%) | 8 (73%) | | | |
| **Lymph node metastasis** | | | | | | |
| N0 | 11 | 4 (36%) | 7 (64%) | **6.427** | **0.040** | **0.209** |
| N1 | 36 | 17 (47%) | 19 (53%) | | | |
| N2 | 13 | 1 (8%) | 12 (92%) | | | |
| **Grade of differentiation** | | | | | | |
| High | 36 | 6 (16%) | 30 (84%) | **13.424** | **0.000** | **0.508** |
| Moderate | 24 | 16 (67%) | 8 (33%) | | | |
| Low | 0 | 0 (0%) | 0 (0%) | | | |

Categorical data were evaluated using the Pearson χ2 test, adjusted by Yates's correction for continuity. The correlation coefficients between protein expression and clinical and pathological parameters were estimated using Spearman correlation.

primary tumor and metastasis were approximated into qualitative ones by dividing the expression level into high and low median 140 and 220, respectively. The expression of MSI2 in primary tumors did not correlate with that of CDH1 in either the primary tumors (p = 1.0) or metastases (p = 0.186) (Table 4). However, the expression level of MSI2 protein correlated positively with that of E-cadherin in metastases (p = 0.027) (Fig 2C).

## MSI2 positively regulates expression of CDH1, ZO1, and TGFb1 in CRC

MSI2 is an RNA-binding protein (RBP) that can regulate translation of downstream target mRNAs. Given our analysis of clinical specimens suggested a connection between MSI2 expression, E-cadherin/CDH1 expression, and tumor grade, particularly in tumor metastases, we analyzed whether MSI2 might directly regulate E-cadherin expression in CRC. We first performed *in silico* analysis to determine if the CDH1 mRNA contained predicted binding

**Table 3. Comparison of frequencies of qualitative indicators with OS and PFS in mCRC patients with synchronous and metachronous metastasis.**

| Clinicopathological parameters | N | Mean OS (months. CI 95%) | Log-rank | p-value | Mean PFS (months. CI 95%) | Log-rank | p-value |
|---|---|---|---|---|---|---|---|
| **Gender** | | | | | | | |
| Male | 32 | 17.0 (13.4–20.6) | 2.479 | 0.115 | 9.0 (7.7–10.3) | 0.471 | 0.493 |
| Female | 28 | 16.0 (12.8–19.2) | | | 8.0 (5.4–10.3) | | |
| **Age range** | | | | | | | |
| ≤65 years | 22 | 17.0 (13.8–20.2) | 0.012 | 0.914 | 8.0 (5.2–10.8) | 0.049 | 0.825 |
| >65 years | 38 | 17.0 (14.0–20.0) | | | 9.0 (7.8–10.2) | | |
| **Type (Localized/Metastasized)** | | | | | | | |
| Synchronous | 31 | 13.5 (9.7–17.3) | **8.963** | **0.003** | 6.0 (3.3–8.7) | **11.395** | **0.001** |
| Metachronous | 29 | 20.0 (13.7–26.3) | | | 11.0 (10.0–12.0) | | |
| **Location of tumor** | | | | | | | |
| Left-sided | 25 | 20.0 (16.8–23.2) | 0.314 | 0.575 | 11.0 (9.0–13.0) | **4.664** | **0.031** |
| Right-sided | 35 | 14.5 (11.6–17.4) | | | 8.0 (5.5–10.5) | | |
| **Stage** | | | | | | | |
| I | 5 | - | **11.592** | **0.009** | 11.0 (8.9–13.1) | **11.591** | **0.009** |
| II | 6 | 17.0 (13.6–20.4) | | | 11.0 (8.7–13.3) | | |
| III | 18 | 19.0 (14.8–23.2) | | | 10.0 (5.8–14.2) | | |
| IV | 31 | 14.0 (10.4–17.6) | | | 6.0 (3.3–8.7) | | |
| **Depth of invasion** | | | | | | | |
| T2 | 7 | - | **6.536** | **0.038** | 11.0 (9.2–12.2) | 2.947 | 0.229 |
| T3 | 42 | 17.0 (14.6–19.4) | | | 9.0 (8.0–10.0) | | |
| T4 | 11 | 11.5 (8.3–14.7) | | | 4.0 (1.8–6.2) | | |
| **Lymph node metastasis** | | | | | | | |
| N0 | 11 | 23.0 (18.0–28.0) | **27.584** | **<0.0001** | 11.0 (10.2–11.8) | **18.338** | **<0.0001** |
| N1 | 36 | 19.0 (15.5–22.5) | | | 9.0 (7.5–10.5) | | |
| N2 | 13 | 11.0 (7.9–14.1) | | | 4.0 (1.7–6.3) | | |
| **Histology** | | | | | | | |
| Adenocarcinoma | 50 | 17.0 (14.7–19.3) | 1.668 | 0.434 | 9.0 (7.3–10.7) | 0.878 | 0.645 |
| Mucinous carcinoma | 8 | 13.0 (10.7–15.3) | | | 7.0 (4.2–9.8) | | |
| Signet-ring cell carcinoma | 1 | - | | | 11.0 | | |
| **Grade of differentiation** | | | | | | | |
| Moderate | 24 | 21.5 (16.7–26.3) | **6.378** | **0.012** | 11.0 (8.6–13.4) | **8.307** | **0.004** |
| High | 36 | 14.5 (11.6–17.4) | | | 8.0 (6.5–9.5) | | |
| **Therapy** | | | | | | | |
| Sole surgical resection | 1 | 6.0 | **13.175** | **<0.0001** | 4.0 | 2.533 | 0.282 |
| + chemotherapy | 55 | 17.0 (14.2–19.8) | | | 9.0 (7.8–10.2) | | |
| + chemoradiation | 4 | 20.0 (5.3–34.7) | | | 11.0 (2.7–19.3) | | |
| **Resection** | | | | | | | |
| R0 | 57 | 17.0 (14.5–19.5) | 0.571 | 0.450 | 9.0 (7.7–10.3) | 1.411 | 0.235 |
| R1+R2 | 3 | 13.5 (11.1–15.9) | | | 8.0 (6.4–9.6) | | |
| **E-cadherin in primary tumor** | | | | | | | |
| Low | 12 | 12.0 (9.5–14.5) | 0.019 | 0.889 | 4.0 (2.6–5.4) | 1.405 | 0.236 |
| High | 14 | 13.0 (9.4–17.6) | | | 8.0 (6.2–9.8) | | |
| **E-cadherin in metastasis** | | | | | | | |
| Low | 12 | 19.0 (13.9–24.1) | **12.291** | **<0.0001** | 11.0 (6.0–16.0) | **8.458** | **0.004** |
| High | 16 | 11.0 (9.5–12.5) | | | 6.0 (4.7–7.3) | | |
| **Musashi 2 in primary tumor** | | | | | | | |

*(Continued)*

**Table 3.** (Continued)

| Clinicopathological parameters | N | Mean OS (months. CI 95%) | Log-rank | p-value | Mean PFS (months. CI 95%) | Log-rank | p-value |
|---|---|---|---|---|---|---|---|
| Low | 21 | 20.0 (15.5–24.5) | 2.849 | 0.091 | 11.5 (9.6–13.4) | **12.627** | **<0.0001** |
| High | 39 | 14.0 (11.3–16.7) | | | 8.0 (6.3–9.7) | | |
| **Musashi 2 in metastasis** | | | | | | | |
| Low | 22 | 24.5 (20.8–28.2) | **12.555** | **<0.0001** | 11.0 (8.7–13.3) | **13.481** | **<0.0001** |
| High | 38 | 13.0 (10.7–15.2) | | | 7.0 (5.0–9.0) | | |

The Kaplan-Meier method was used to construct survival curves, as well as to calculate the median of survival. Differences between groups were established using a log-rank test.

motifs for MSI2 (Fig 3A). For this, we looked for motifs suggested by Wang et al [25], focusing on occurrence of longer (7 or 8 nucleotide) binding consensus sites, as these were less likely to occur by chance (Fig 3A, S3 Table). We performed similar analysis for two other genes known to be relevant to tumor grade in CRC: the tight junction-associated protein ZO-1/TJP1, and a regulator of epithelial-mesenchymal transition, TGFβ1 [25]. This analysis identified 3

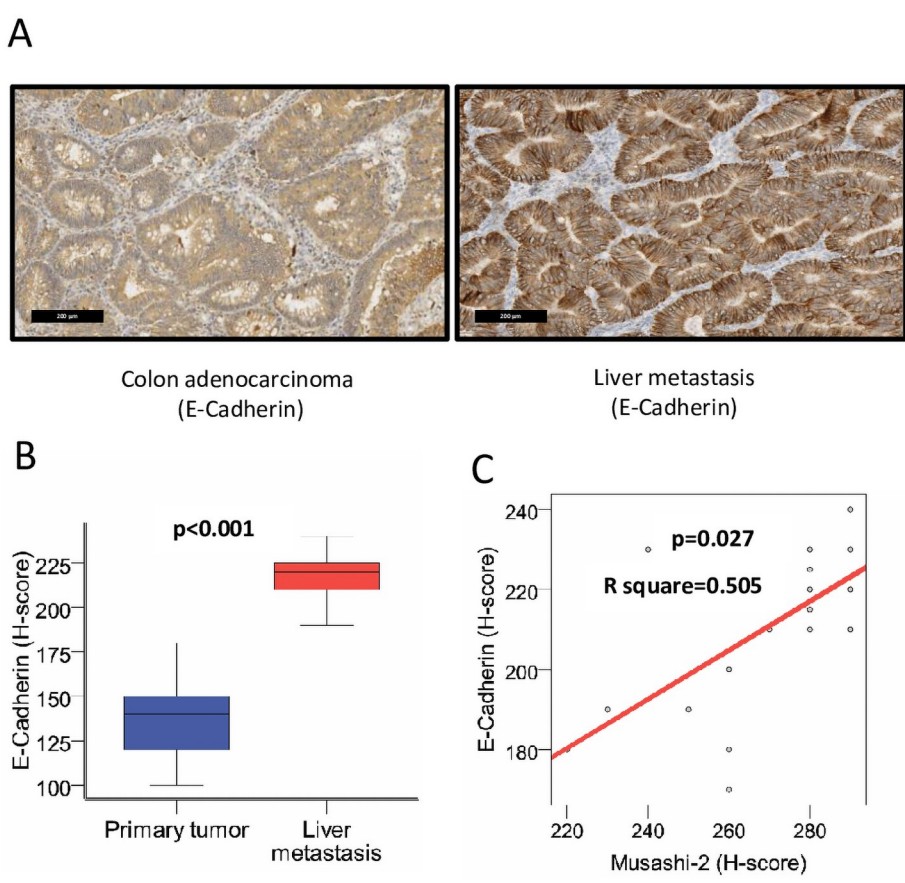

**Fig 2. Expression of E-cadherin/CDH1 in primary CRC and liver metastases; correlation with MSI2 expression.** (**A**) Representative IHC expression of E-cadherin/CDH1 in primary CRC and liver metastasis from individual patient. (**B**) H-score for E-cadherin/CDH1 in primary tumor and liver metastasis (n = 28). (**C**) Correlation of H-score for MSI2 and E-cadherin/CDH1 in liver metastases ($R^2$ = 0.505, p = 0.027).

**Table 4. Relationship between MSI2 and E-cadherin in CRC.**

| Clinicopathological parameters | Total Number of patients | MSI2 protein expression | | χ2 | p-value | r |
|---|---|---|---|---|---|---|
| | | Low (n. %) | High (n. %) | | | |
| Expression level in primary tumor | | | | | | |
| E-cadherin in primary tumor | | | | | | |
| Low | 12 | 3 (25%) | 9 (75%) | 0.000 | 1.000 | -0.040 |
| High | 16 | 5 (32%) | 11 (68%) | | | |
| E-cadherin in metastasis | | | | | | |
| Low | 12 | 5 (42%) | 7 (58%) | 1.750 | 0.186 | 0.333 |
| High | 16 | 2 (12%) | 14 (88%) | | | |
| Expression level in metastasis | | | | | | |
| E-cadherin in primary tumor | | | | | | |
| Low | 12 | 1 (8%) | 11 (92%) | 0.650 | 0.420 | -0.256 |
| High | 16 | 5 (32%) | 11 (68%) | | | |
| E-cadherin in metastasis | | | | | | |
| Low | 12 | 6 (50%) | 6 (50%) | **4.861** | **0.027** | **0.500** |
| High | 16 | 1 (6%) | 15 (94%) | | | |

Categorical data were evaluated using the Pearson χ2 test, adjusted by Yates's correction for continuity. The correlation coefficients between protein expression and clinical and pathological parameters were estimated using Spearman correlation.

predicted MSI2 binding sites in the 3' untranslated region (UTR) of the *CDH1* mRNA, 4 in the *ZO-1/TJP1* UTR, and 1 in the *TGFβ1* UTR.

As a first approach to exploring the suggested relationships, we assessed whether MSI2 expression correlated with expression of CDH1, ZO-1/TJCP1, or TGFβ1 mRNA or protein, using data from The Cancer Genome Atlas (TCGA) Research Network. Based on the idea that MSI2 regulates protein translation of its targets, we would predict correlated expression at the protein level, but no correlation, or much lower correlation, at the mRNA level. In a data set of 83 CRC cases for which matching RNA and protein data was available, we found no correlation between MSI2 and TGFβ1 mRNA expression, and we only observed modest correlation between MSI2 and ZO1/CDH1 mRNA expression (Fig 3B). However, we observed significantly correlated protein expression of MSI2 with TGFβ1, ZO-1 and CDH1 protein, consistent with a role of MSI2 in regulating primarily mRNA translation for these genes (Fig 3C).

Next, we used RT-PCR and western analysis to directly determine if MSI2 depletion reduced the protein, but not mRNA, expression of these candidate targets. Use of two independent shRNAs (-sh1,-sh2) to deplete MSI2, versus empty vector, in two human CRC cell lines (RKO and HCT116), significantly decreased TGFβ1, CDH1, and ZO-1 protein expression in both cell lines, while not affecting mRNA levels for these proteins (Fig 4A–4D). These results were consistent with a primary role of MSI2 as an RBP in regulating protein translation rather than regulating expression of mRNA.

## MSI2 supports viability and clonal growth of CRC cell lines

Finally, we assessed whether depletion of MSI2 affected the viability and clonogenic capacities of CRC cell lines, again using shRNA to deplete MSI2. Three days after MSI2 depletion, use of a WST-1/CelltiterBlue assay indicated that MSI2 depletion significantly decreased viability of both the RKO and HCT116 human CRC cell lines versus cells bearing negative control shRNA (Fig 4E and 4F). Similarly, MSI2 depletion also significantly suppressed colony formation

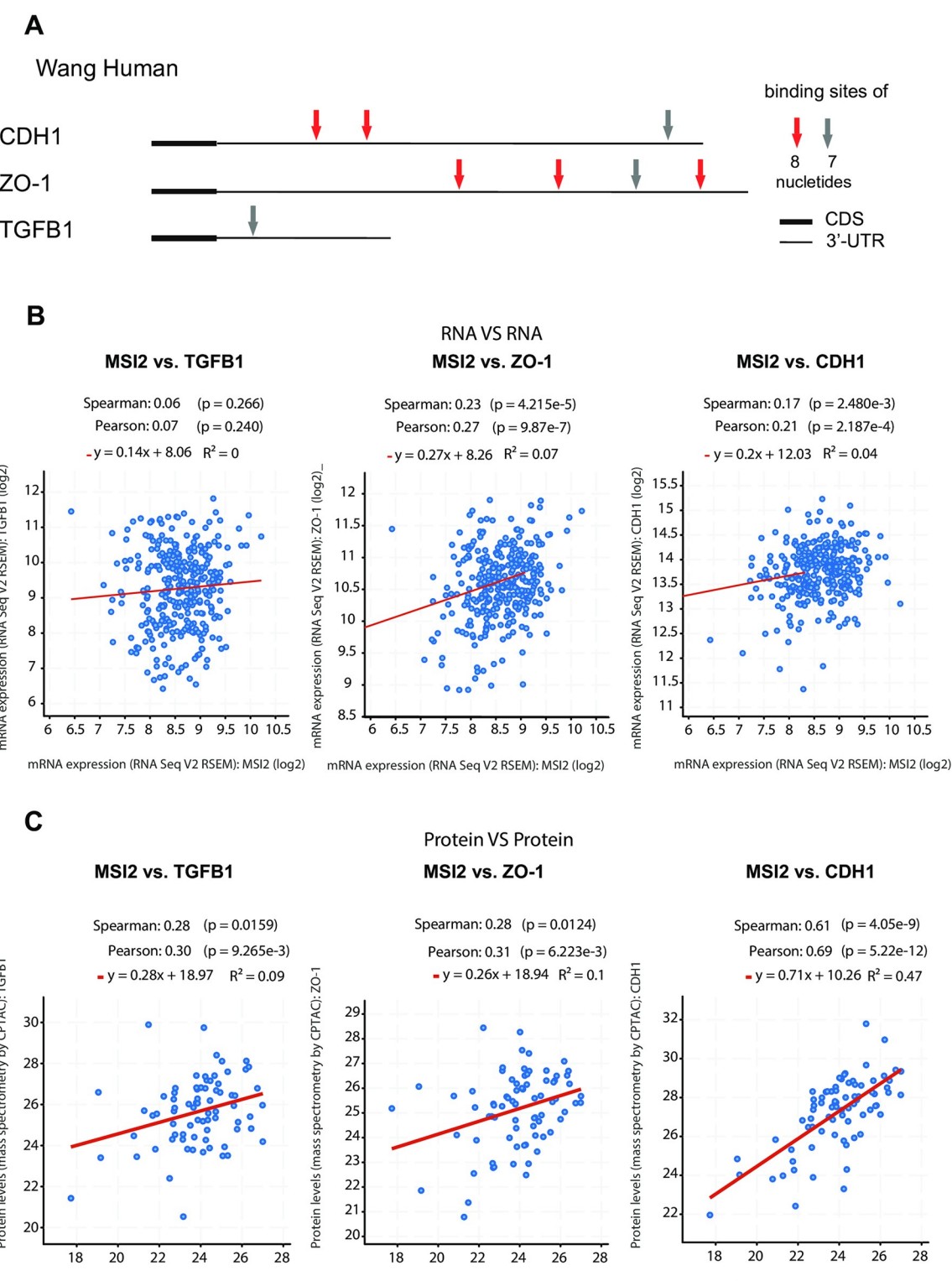

**Fig 3. *In silico* analysis of MSI2 gene regulation in CRC.** (**A**) Location of predicted binding sites for human MSI2 protein defined from study by Wang *et al*. [25], using eight MSI2 binding motifs with the highest p values). Coding sequences (CDS) are represented by thick lines; 3' untranslated regions (3'UTR) by thin line. Shorter consensus sequences are not indicated. (**B, C**) Correlation analysis for expression between MSI2 and CDH1, TGFβ1, ZO1, based on 203 CRC mRNA specimens (**B**) or 83 CRC protein specimens (**C**), based on data downloaded from www.cbioportal.org (TCGA, Firehouse legacy, Date: January 22 2020).

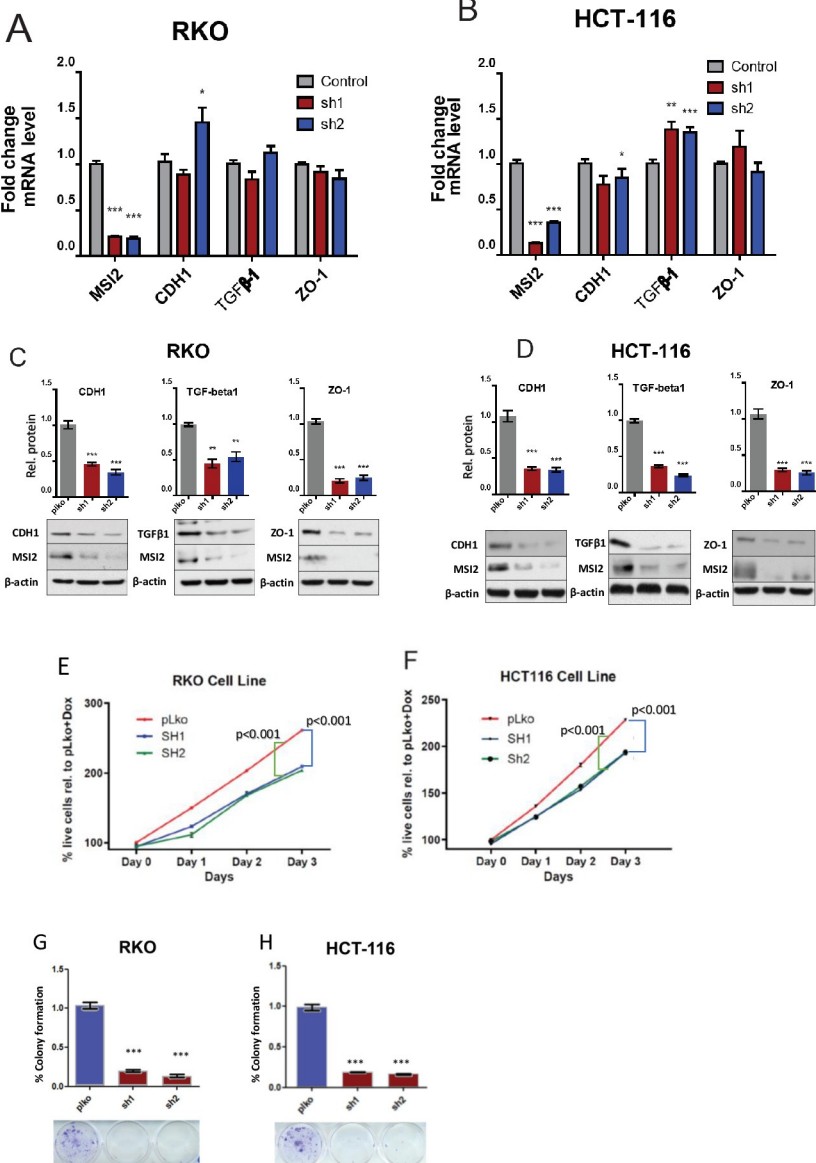

**Fig 4. In vitro assessment of MSI2 depletion in CRC cell lines. (A-D)** Averaged values for qRT-PCR (**A, B**) and Western blot (**C, D**) for MSI2, CDH1, TGFβ1, and ZO1 in control and MSI2-depleted RKO and HCT116 cell lines. (**E-H**) Cell viability (**E, F**) and clonogenic capacity (**G, H**) of control or MSI2-depleted RKO and HCT116 CRC cell lines. *, $p < 0.05$; **, $p < 0.01$; ***, $p < 0.001$, in each case in reference to control-depleted cells; non-significant comparisons are not indicated.

capacity of both cell lines in a clonogenic assay, reducing staining intensity by more than 80% (Fig 4G and 4H).

## Discussion

In this study, we show that MSI2 expression in the colon is elevated by stages during malignant transformation. MSI2 expression is initially elevated in pre-cancerous tubulovillus adenomas (polyps), further elevated in primary tumors, and then dramatically increased in distant

metastases compared to normal mucosa and primary tumor. We find that regardless of tumor stage, high levels of MSI2 in primary tumors were associated with shorter PFS. Elevated MSI2 expression in liver metastasis linked to both poor PFS and OS, and intriguingly, were associated with high expression of E-cadherin/CDH1, a marker of epithelial identity. Pursuing this last observation in analysis of public datasets, we found that MSI2 protein expression correlated with that of CDH1 and other proteins implicated in control of epithelial versus mesenchymal identity. We also demonstrated directly that depletion of MSI2 in vitro reduced protein but not RNA expression of CDH1, ZO-1, and TGFβ1, and that the 3'UTRs in the mRNAs in these genes contained candidate MSI2 binding sites. We further showed MSI2 depletion reduced viability and clonogenic capacity of CRC cell models.

Overall, these data suggested MSI2 has a complex action in CRC that varies dependent on tumor stage. MSI2 support of the expression of proteins associated with both epithelial (E-cadherin and ZO-1) and mesenchymal (TGFβ1) identity is consistent with a role of MSI2 reported previously in non-small cell lung cancer (NSCLC), in which MSI2 overexpression induced a mixed differentiation phenotype [9]. In that study, MSI2 was identified as a metastatic driver that supported the protein expression of the TGFβ receptor TGFBR1, and TGFβ effector SMAD3, suggesting MSI2 was required for EMT; however, MSI2 depletion reduced CDH1 expression in NSCLC cell cultures, indicating a role in supporting epithelial identity [9]. A growing number of studies have emphasized the presence of quasi-mesenchymal cells in aggressive tumors [28]; these cells, which are marked by extreme lineage plasticity, are associated with poor response to therapy and poor survival. In CRC progression, malignancy is promoted by loss of differentiation in pre-metastatic cells, but increased differentiation in cancer cells after extravasation, as this is necessary to form rapidly proliferating tumors in the metastatic niche. We hypothesize that the specific correlation between MSI2 and CDH1 expression in liver metastases may indicate a specific role for MSI2 in supporting CDH1 expression in the metastatic setting. In contrast, MSI2 may be more active in regulating TGFβ1 expression and supporting proliferation in primary tumors, and may also contribute to the proliferation of pre-cancerous polyps.

Several prior studies have identified roles for MSI2 as an oncogenic driver in multiple types of cancer [14,15,18,29]. Particularly pertinent to this study, work in murine CRC models have shown a direct oncogenic role of MSI2, as well as the importance of overexpression of MSI2 and APC gene loss [15]. In the TRE-MSI2 mouse model, overexpression of MSI2 in the intestinal epithelium was associated with high crypt proliferation and crypt fission, a block in differentiation, and development of specific morbidities that were similar to acute loss of APC [25]. MSI2 overexpression in these mice provoked rapid and substantial shifts in the intestinal epithelial transcriptome, with gene set enrichment analysis of these changes showing that the APC-loss gene signature was among the most significantly enriched [25]. In addition, overexpressed MSI2 plays an oncogenic role in myeloid leukemia, with elevated expression of MSI2 is linked with poor survival in leukemia patients [13]. Other tumor types in which MSI2 overexpression has been linked to increased aggressiveness include hepatocellular carcinoma and gastric cancer [11].

In summary, our data indicate that elevated expression of the MSI2 protein is a prognostic biomarker for poor prognosis both in pre-cancerous conditions and at multiple stages of CRC. Further, MSI2 actively promotes CRC growth, and is potentially a promising therapeutic target, with efforts currently ongoing to identify drugs targeting MSI2 RNA-binding activity [11]. This is of potential clinical value, as in many cases, identification of specific mutations or epigenetic changes occurring early in tumor formation has been shown to help stratify patients for clinical management, improving therapeutic outcomes [23,30]. As one example, a recently proposed classification of four consensus molecular subtypes (CMS) can be used as a basis for studying the sensitivity of various molecular subtypes to therapy; studies using this

classification determined that irinotecan is particularly effective in CMS4 patients [27]. As another example, the presence of BRAF mutations in tumors from stage III-IV CRC patients is associated with decreased survival [31,32]. Additional retrospective studies and a post hoc analysis of a prospective clinical trial have indicated that abnormal DNA methylation can affect prognosis in colon cancer, with CpG Island Methylator Phenotype positive (CIMP+) colon cancers predicting poor survival compared to CIMP negative cancers [33,34]. As a regulator of protein translation, MSI2 is positioned to influence expression of numerous cancer targets in addition to those identified here; based on the work presented here, further study in CRC is strongly merited.

## Supporting information

**S1 Table. List of cell line derivatives used in the study.** For human cell lines, the lentiviral vectors Tet-pLKO-puro (Addgene, Plasmid #21915) was used for inducible expression of shRNAs.
(DOCX)

**S2 Table. DNA oligonucleotides used for construction of shRNA vectors.** Table lists of single stranded DNA (ssDNA) oligos used for generation of Tet-pLKO-puro vectors expressing specific shRNAs, which used for lentiviral infection and selection of stable cell lines. MSI2 targeting sequences are underscored.
(DOCX)

**S3 Table. Consensus MSI2 binding site sequences, defined by Wang et al.**
(XLSX)

**S4 Table. Primers used for RT-PCR to quantify gene expression.** List of primers used for SYBR Green assay, and TaqMan gene expression assay, used for qRT-PCR analysis of gene expression.
(DOCX)

**S1 File. WB Musashi CRC project.**
(PDF)

## Author Contributions

**Conceptualization:** Leonid Kharin, Elena Frantsiyants, Oleg Kit, Yanis Boumber.

**Data curation:** Leonid Kharin, Nikolay Karnaukhov, Mark Voloshin.

**Formal analysis:** Leonid Kharin, Mark Voloshin.

**Funding acquisition:** Erica Golemis, Yanis Boumber.

**Investigation:** Leonid Kharin, Igor Bychkov, Nikolay Karnaukhov, Mark Voloshin, Rushaniya Fazliyeva.

**Methodology:** Leonid Kharin, Igor Bychkov, Alexander Deneka, Elena Frantsiyants, Erica Golemis, Yanis Boumber.

**Project administration:** Leonid Kharin, Mark Voloshin, Elena Frantsiyants, Erica Golemis, Yanis Boumber.

**Resources:** Erica Golemis, Yanis Boumber.

**Software:** Leonid Kharin, Igor Bychkov, Mark Voloshin.

**Supervision:** Oleg Kit, Erica Golemis, Yanis Boumber.

**Validation:** Leonid Kharin.

**Visualization:** Leonid Kharin, Mark Voloshin, Erica Golemis, Yanis Boumber.

**Writing – original draft:** Leonid Kharin, Erica Golemis, Yanis Boumber.

**Writing – review & editing:** Leonid Kharin, Igor Bychkov, Mark Voloshin, Erica Golemis, Yanis Boumber.

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
