## [Decision Letter · Decision Letter 0]

24 Aug 2020

PONE-D-20-17649

Prognostic role and biologic features of Musashi-2 expression in colon polyps and during colorectal cancer progression

PLOS ONE

Dear Dr. Kharin,

Thank you for submitting your manuscript to PLOS ONE. After careful consideration, we feel that it has merit but does not fully meet PLOS ONE’s publication criteria as it currently stands. Therefore, we invite you to submit a revised version of the manuscript that addresses the points raised during the review process.

Please submit your revised manuscript within 45 days of this decision. If you will need more time than this to complete your revisions, please reply to this message or contact the journal office at plosone@plos.org. Please include the following items when submitting your revised manuscript:

We look forward to receiving your revised manuscript.

Kind regards,

Punita Dhawan

Academic Editor

PLOS ONE

Journal Requirements:

2.Thank you for including your ethics statement: 'IRB of National Medical Research Center of Oncology (Rostov-on-Don, Russia) (IRB No. 6, 02/03/2018).'

(a) Please amend your current ethics statement to include the full name of the ethics committee/institutional review board(s) that approved your specific study.  

(b) Please amend your current ethics statement to specifically state that the IRB approved your study.

(c) Once you have amended this/these statement(s) in the Methods section of the manuscript, please add the same text to the “Ethics Statement” field of the submission form (via “Edit Submission”).

3. In the ethics statement in the manuscript and in the online submission form, please provide additional information about the patient samples used in your retrospective study, including: a) whether all samples were fully anonymized before you accessed them; b) the date range (month and year) during which patients' samples were accessed; c) the date range (month and year) during which patients whose samples were selected for this study sought treatment; and d) the source of the samples analyzed in this work (e.g. hospital, institution or medical center name).

4. Please provide additional information about the HEK293T cells used in this work, including source, culture conditions and any quality control testing procedures (authentication and mycoplasma testing). For more information, please see http://journals.plos.org/plosone/s/submission-guidelines#loc-cell-lines.

5. At this time, we ask that you please provide scale bars on the microscopy images presented in Figure 1 and 2 and refer to the scale bar in the corresponding Figure legend.

6. To comply with PLOS ONE submission guidelines, in your Methods section, please provide additional information regarding your statistical analyses for the in vitro cell line studies. For more information on PLOS ONE's expectations for statistical reporting, please see https://journals.plos.org/plosone/s/submission-guidelines.#loc-statistical-reporting.

7.PLOS ONE now requires that authors provide the original uncropped and unadjusted images underlying all blot or gel results reported in a submission’s figures or Supporting Information files. This policy and the journal’s other requirements for blot/gel reporting and figure preparation are described in detail at https://journals.plos.org/plosone/s/figures#loc-blot-and-gel-reporting-requirements and https://journals.plos.org/plosone/s/figures#loc-preparing-figures-from-image-files. When you submit your revised manuscript, please ensure that your figures adhere fully to these guidelines and provide the original underlying images for all blot or gel data reported in your submission. See the following link for instructions on providing the original image data: https://journals.plos.org/plosone/s/figures#loc-original-images-for-blots-and-gels.

8. Please upload a copy of Supp. Table S1., Supp. Table S2., Supp. Table S3., Supp. Table S4.,  which you refer to in your text on page 6 and 7.

Reviewers' comments:

Reviewer's Responses to Questions

**Comments to the Author**

1. Is the manuscript technically sound, and do the data support the conclusions?

Reviewer #1: Yes

Reviewer #2: Yes

2. Has the statistical analysis been performed appropriately and rigorously? 

Reviewer #1: Yes

Reviewer #2: Yes

3. Have the authors made all data underlying the findings in their manuscript fully available?

Reviewer #1: Yes

Reviewer #2: Yes

4. Is the manuscript presented in an intelligible fashion and written in standard English?

Reviewer #1: Yes

Reviewer #2: Yes

5. Review Comments to the Author

Reviewer #1: RNA-binding protein Musashi-2 (MSI2) controls the translation of proteins that support stem cell identity and lineage determination and is associated with progression in some cancers. The present study explore the prognostic significance of MSI2 in CRC polyps and colon cancer. Its well-planned study that explicitly signifies the role of MSI2 in CRC. However, the data on its role in initiation is still unclear. In order to clarify its role and prognostic significance, it will be interesting to see MSI2 expression alteration in hyperplastic, and serrated polyp types. How microsatellite instability impact MSI2 expression and does its differential expression is associated with mutational status of tumor.

Minor Suggestion: A high level of MSI2 expression in the metastasis was also associated with lymph node positivity (median values with quartiles 270 [250;280] for N0 versus 265 [242;280] for N1 versus 280 [275;290] for N2; average values, 268 versus 262 versus 281; p=0.04), and with higher differentiation grade of the primary tumor (p = 0.000). Clearly mention the p-value observed for the study.

Reviewer #2: The manuscript entitled "Prognostic role and biologic features of Musashi-2 expression in colon polyps and during colorectal cancer progression" by Leonid Kharin et al. Here authors, addressed the critical role of Musashi-2 (MSI2) in colorectal cancer and metastasis. The idea of this article is exciting. The author showed a stage-wise increase in the expression of MSI2. Also, it showed the critical role of MSI cell proliferation and growth. However, colon cancer-promoting role of MSI2 is already reported. The techniques or methodology used is up to date. However, the following points need to be addressed.

1. In one section of the results, the author stated that the expression of MSI2 in primary tumors did not correlate with CDH1 expression. However, in another section using the TCGA data set, the author showed a positive correlation between MSI2 protein expression and CDH1 protein. This ambiguity needs to be addressed.

2. The whole idea of this manuscript to show the significance of MSI2 in prognosis colon cancer progression. It would be better to use regression and ROC analysis methods to show the prognostic significance of MSI2 in predicting disease progression.

3. Figure1A: MSI2 expression (representative IHC images) in adenocarcinoma and liver metastasis is not clear and hard to interpret. Higher magnification insert would help in compression between them.

4. Figure1D: Author showed MSI2 expression comparison between grade 3 with grade1 & 2. It would be better to show compressions among normal and all the grades with representative IHC images of MSI2.

5. Figure4G-H: typo error in Y axis, % colony formation instead of % cell viability.

6. PLOS authors have the option to publish the peer review history of their article (what does this mean?). If published, this will include your full peer review and any attached files.

Reviewer #1: No

Reviewer #2: No

---

## [Author Response · Author response to Decision Letter 0]

29 Dec 2020

December 19, 2020

Punita Dhawan, PhD 

Editorial Office

PLoS One

Dear Dr. Dhawan,

We here submit our revised original research article, ““"Prognostic role and biologic features of Musashi-2 expression in colon polyps and during colorectal cancer progression"” (PONE-D-20-17649), to be considered for publication in PLoS One. We greatly appreciate the expert comments and suggestions from the reviewers. We have addressed most of the points raised by the reviewers, and the appropriate changes have been made to the manuscript, as detailed below. A small number of points could not be addressed due to technical reasons which we explain in the following point-by-point response letter and additional data attached. We hope that you agree that the revised manuscript is significantly improved and is now acceptable for publication.

Reviewer #1: RNA-binding protein Musashi-2 (MSI2) controls the translation of proteins that support stem cell identity and lineage determination and is associated with progression in some cancers. The present study explore the prognostic significance of MSI2 in CRC polyps and colon cancer. It is a well-planned study that explicitly signifies the role of MSI2 in CRC. 

Response: We thank the Reviewer for the positive assessment of the study.

Point #1. …The data on its role in initiation is still unclear. In order to clarify its role and prognostic significance, it will be interesting to see MSI2 expression alteration in hyperplastic, and serrated polyp types. 

Response #1: We have added new images showing the expression of MSI2 in 1 hyperplastic polyp and 1 serrated adenoma (revised Figure 1). These data suggest that MSI2 is not expressed in these early stage lesions. However, the low number of samples available does not allow us to draw statistically significant conclusions, so we are cautious in discussing these results.

Point #2. How microsatellite instability impact MSI2 expression and does its differential expression is associated with mutational status of tumor.

Response #2: This is an important point. In our study, we analyzed 39 tumors that were microsatellite stable, and 5 tumors that had microsatellite instability (numbers now indicated in revised Table 1). We found no impact of microsatellite instability status on MSI2 expression level (p=0.932). We have added a sentence to the results to make this point.

Minor Suggestion: A high level of MSI2 expression in the metastasis was also associated with lymph node positivity (median values with quartiles 270 [250;280] for N0 versus 265 [242;280] for N1 versus 280 [275;290] for N2; average values, 268 versus 262 versus 281; p=0.04), and with higher differentiation grade of the primary tumor (p = 0.000). Clearly mention the p-value observed for the study.

Response for suggestion: We have clarified the description of the results with associated p-values in the main text of the manuscript. 

Reviewer #2: The manuscript entitled "Prognostic role and biologic features of Musashi-2 expression in colon polyps and during colorectal cancer progression" by Leonid Kharin et al. Here authors, addressed the critical role of Musashi-2 (MSI2) in colorectal cancer and metastasis. The idea of this article is exciting. The author showed a stage-wise increase in the expression of MSI2. Also, it showed the critical role of MSI cell proliferation and growth. However, colon cancer-promoting role of MSI2 is already reported. The techniques or methodology used is up to date. However, the following points need to be addressed.

Response: We thank the Reviewer for the positive assessment of the study.

Point #1. In one section of the results, the author stated that the expression of MSI2 in primary tumors did not correlate with CDH1 expression. However, in another section using the TCGA data set, the author showed a positive correlation between MSI2 protein expression and CDH1 protein. This ambiguity needs to be addressed.

Response #1: We apologize for the apparent contradiction of these data. Due to limited tissue for both MSI2 and CDH1 staining, we were only able to stain colon tumors from 28 patients in our study using IHC vs. 83 patients in TCGA, providing more statistical power to the TCGA data set. Furthermore, besides difference in the number of patients, different methods were used; IHC in our study vs. mass spectrometry in TCGA. Based on all of these factors, it is clear that additional studies should be done to assess the functional relationship of MSI2 and CDH1 expression. This point was added to the Discussion section. 

Point #2. The whole idea of this manuscript to show the significance of MSI2 in prognosis colon cancer progression. It would be better to use regression and ROC analysis methods to show the prognostic significance of MSI2 in predicting disease progression.

Response #2: Thank you for your comment. We applied Cox regression analysis and ROC analysis to the data; however, these methods sdid not show any significance in relationship of MSI2 expression in primary CRC and prognosis in patients. We added this statement to the manuscript Results section.

Point #3. Figure 1A: MSI2 expression (representative IHC images) in adenocarcinoma and liver metastasis is not clear and hard to interpret. Higher magnification insert would help in compression between them.

Response #3: We apologize for the quality of the initially submitted images. We have updated Figure 1A, and replaced old images with the new images in this Figure.

Point #4. Figure 1D: Author showed MSI2 expression comparison between grade 3 with grade1 & 2. It would be better to show comparisons among normal and all the grades with representative IHC images of MSI2.

Response #4: We added new images with Grade 1 – 3 tumors (MSI2 IHC staining). 

Point #5. Figure 4G-H: typo error in Y axis, % colony formation instead of % cell viability.

Response #5: We apologize for this typo. We updated the Y-axis in Figures 4G and 4H. 

 We once again thank the Editor and the Reviewers for a fair review, and hope that the improved manuscript would be suitable for PLoS One.

Sincerely yours,

Yanis Boumber, MD, PhD

Leonid Khaein , MD

---

## [Decision Letter · Decision Letter 1]

11 May 2021

Prognostic role and biologic features of Musashi-2 expression in colon polyps and during colorectal cancer progression

PONE-D-20-17649R1

Dear Dr. Kharin,

We’re pleased to inform you that your manuscript has been judged scientifically suitable for publication and will be formally accepted for publication once it meets all outstanding technical requirements.

Kind regards,

Punita Dhawan

Academic Editor

PLOS ONE

Additional Editor Comments (optional):

Reviewers' comments:

Reviewer's Responses to Questions

**Comments to the Author**

1. If the authors have adequately addressed your comments raised in a previous round of review and you feel that this manuscript is now acceptable for publication, you may indicate that here to bypass the “Comments to the Author” section, enter your conflict of interest statement in the “Confidential to Editor” section, and submit your "Accept" recommendation.

Reviewer #2: All comments have been addressed

Reviewer #3: All comments have been addressed

2. Is the manuscript technically sound, and do the data support the conclusions?

Reviewer #2: Yes

Reviewer #3: Yes

3. Has the statistical analysis been performed appropriately and rigorously? 

Reviewer #2: Yes

Reviewer #3: Yes

4. Have the authors made all data underlying the findings in their manuscript fully available?

Reviewer #2: Yes

Reviewer #3: Yes

5. Is the manuscript presented in an intelligible fashion and written in standard English?

Reviewer #2: Yes

Reviewer #3: Yes

6. Review Comments to the Author

Reviewer #2: The manuscript entitled "Prognostic role and biologic features of Musashi-2 expression in colon polyps and during colorectal cancer progression" is now looks satisfactory and ready for acceptance.

Reviewer #3: Authors have made revisions that are sufficient to address the concerns. Thanks for all the revisions.

7. PLOS authors have the option to publish the peer review history of their article (what does this mean?). If published, this will include your full peer review and any attached files.

Reviewer #2: No

Reviewer #3: No

---

## [Editor Report · Acceptance letter]

25 Jun 2021

PONE-D-20-17649R1 

Prognostic role and biologic features of Musashi-2 expression in colon polyps and during colorectal cancer progression 

Dear Dr. Kharin:

I'm pleased to inform you that your manuscript has been deemed suitable for publication in PLOS ONE. Congratulations! Your manuscript is now with our production department. 

Kind regards, 

on behalf of

Dr. Punita Dhawan 

Academic Editor

PLOS ONE